# Statistical methods for predicting the presence of Salmonella Typhi in wastewater samples at Asante Akyem Agogo, Ghana

Sampson Twumasi-Ankrah[1]*, Michael Owusu[2], Michael Owusu-Ansah[3], Seidu Amenyaglo[4], Caleb Osei-Wusu Sarfo[4], Eric Darko[5], Portia Okyere Boakye[4], Christopher B. Uzzell[6], Isobel M. Blake[6], Nicholas C. Grassly[6], Yaw Adu-Sarkodie[5], Ellis Owusu-Dabo[7]

**1** Department of Statistics and Actuarial Science, Kwame Nkrumah University of Science and Technology, Kumasi, Ghana, **2** Department of Medical Diagnostics, Kwame Nkrumah University of Science and Technology, Kumasi, Ghana, **3** Department of Community Health, Kwame Nkrumah University of Science and Technology, Kumasi, Ghana, **4** KNUST-IVI Collaborative Centre, Agogo, Ashanti Region, Ghana, **5** Department of Clinical Microbiology, Kwame Nkrumah University of Science and Technology, Kumasi, Ghana, **6** Department of Infectious Disease Epidemiology, Imperial College London, London, United Kingdom, **7** School of Public Health, Kwame Nkrumah University of Science and Technology, Kumasi, Ghana

* stankrah.cos@knust.edu.gh

## Abstract

### Background

Monitoring wastewater is vital for tracking typhoid fever in endemic areas. This study evaluated the performance of both spatial and non-spatial models in predicting Salmonella Typhi detection in wastewater from the Asante Akim North district in Ghana and identified key environmental risk factors.

### Methods

We collected wastewater samples of Moore swabs at 40 sites across Agogo, Juansa, Hwidiem, and Domeabra over a period of 27 months. Multiplex PCR was used to detect Salmonella Typhi, focusing on the ttr, tviB, and staG genes. An Aquaprobe AP-2000 was also used to measure different physicochemical factors, such as pH, temperature, dissolved oxygen, and salinity. Three non-spatial models, namely Generalized Estimating Equations (Logistic), Mixed-Effects Models, and Random Forest, as well as four spatial models, including Bayesian Generalized Additive Models (GAM) and Spatial Generalized Linear Mixed Models (GLMM), were fitted to the wastewater dataset. Model fitting was done using 5-fold cross-validation, stratified by site. Model performance was evaluated using accuracy, sensitivity, and specificity. We also used SHapley Additive exPlanations (SHAP) analysis to find the most important predictors.

**Data availability statement:** The dataset for this manuscript is available on OSF under S_Typhi. https://doi.org/10.17605/OSF.IO/WGPUM (TWUMASI-ANKRAH, 2025).

**Funding:** This work was supported by the Bill and Melinda Gates Foundation [INV-002381 to N.C.G.]. The funders had no role in study design, data collection and analysis, decision to publish, or preparation of the manuscript.

**Competing interests:** The authors have declared that no competing interests exist.

## Findings

In general, 44.13% of the samples tested positive for S. Typhi. Detection was much higher during wet seasons (50.17% vs. 35.11%; $p < 0.001$), with fast flows (64.45%), and in channels that were 1–2 meters wide (58.70%). Positive samples had relatively higher pH (7.46 vs. 7.40; $p < 0.001$), dissolved oxygen (46.97% vs. 36.77%; $p < 0.001$), and rainfall (3.92mm vs. 3.30mm; $p = 0.022$). In comparing both non-spatial and spatial models, the non-spatial Random Forest model demonstrated the highest performance with an accuracy of 0.993, sensitivity of 0.997, and specificity of 0.989. In the SHAP analysis of the preferred non-spatial random forest model, it was found that pH, season, dissolved oxygen, positivity from the previous month, and channel width were identified as the best predictors.

## Conclusion

S. Typhi detection is influenced by wastewater physicochemical properties, with pH, seasonal rainfall, and hydraulic conditions being the most significant. The non-spatial random forest model significantly outperforms both spatial and other non-spatial statistical methods.

## Author summary

Typhoid fever remains a significant public health concern in resource-limited areas with inadequate water and sanitation infrastructure. Monitoring Salmonella Typhi in wastewater provides a cost-effective method for tracking community transmission, particularly in regions where clinical surveillance is limited. In this study, we analyzed wastewater samples collected over 27 months from 40 sites in the Asante Akim North district of Ghana. We used statistical and machine learning models to predict the presence of S. Typhi and to identify key environmental factors that influence its detection. Our results indicate that pH levels, seasonality, dissolved oxygen, and channel width significantly affect detection rates. The non-spatial Random Forest model outperformed both spatial and traditional models, achieving an accuracy of 99.3%. These findings highlight the potential of combining wastewater-based surveillance with machine learning techniques to improve predictions of typhoid outbreaks and inform targeted public health interventions in endemic areas.

## Introduction

Typhoid fever, which is caused by Salmonella enterica serovar Typhi (S. Typhi), is still a big public health problem in places with few resources, where poor water and sanitation systems make it easy for the disease to spread through faeces [1–3]. Every year, there are between 11 and 21 million cases and 135,000–230,000 deaths

around the world [4]. In 2017 alone, sub-Saharan Africa had 1.2 million cases and 29,000 deaths [5]. Typhoid fever is one of the top 20 causes of outpatient morbidity in Ghana, with rates of 112–170 cases per 100,000 person-years. It affects children under 15 more than adults [6,7].

Wastewater surveillance is an inexpensive way to keep track of the spread of typhoid in places where clinical testing is limited. Detecting S. Typhi in environmental samples can aid in identifying community outbreaks and hotspots, rather than solely depending on passive clinical reporting. However, to transform wastewater data into useful insights, we need robust predictive models that consider how environmental factors and time evolve.

Statistical models have been used to predict the risk of typhoid [8–11], but gaps still exist in making these tools better for wastewater surveillance. Previous research in Vellore and Blantyre used mixed-effect models to connect the detection of S. Typhi to environmental factors [12], but there are not many studies that compare different modelling methods, especially in sub-Saharan Africa. For instance, [13] showed that mixed-effects and machine learning can be useful for enterovirus surveillance, but no one has yet looked at how well they work for S. Typhi in wastewater in a systematic way.

This study fills in these gaps by (1) Using Generalized Estimating Equations (GEE), mixed-effects models, and random forest techniques, both in spatial and non-spatial contexts, to predict the presence of S. Typhi in wastewater from the Asante Akim North district of Ghana; and (2) using SHapley Additive exPlanations (SHAP) analysis to find important physicochemical properties of wastewater (like pH, dissolved oxygen, and seasonality) to help with targeted interventions. This research moves the use of wastewater surveillance forward as a public health tool by providing a way to monitor typhoid in places with few resources.

## Materials and methods

### Ethics statement

We obtained ethics approval for this study from the Committee on Human Research Publication and Ethics (CHRPE) of the School of Medical Sciences, Kwame Nkrumah University of Science and Technology (KNUST), Kumasi, Ghana.

For the environmental surveillance program, we did not seek or request informed consent because the samples were wastewater and did not involve human subjects.

This study is a secondary analysis of data that has already been published. It is different from our previous work [14] because it uses a bigger sample size that was collected over 27 months. In contrast to our previous study, this analysis employs statistical modeling techniques to predict the detection of S. typhi and to identify key physicochemical parameters associated with its presence. Each of these steps is elaborated on in the following sections.

### Study area and site selection

The research was conducted in four towns namely, Agogo, Juansa, Hwidiem, and Domeabra, within the Asante Akim North district of Ghana. This district spans an area of 1,099.7 km², with most residents residing in peri-urban areas. A robust Demographic Surveillance System (DSS) is established in the district, wherein households and structures are digitally mapped via GIS for easy identification and tracking. According to a census conducted in 2019, the study area has an estimated population of 109,840.

Based on our study protocol published and implemented by [14], 40 sites were chosen and validated. Each site collected monthly repeat samples for 27 months. The spatial and temporal distributions are provided in another article [14].

### Study design

This was a longitudinal study that was conducted over a 27-month period from June 2022 to September 2024. Samples were collected from 40 sites located in peri-urban areas of the Asante Akim North district of the Ashanti region.

## Sampling collection and laboratory analysis

This constitutes the collection of wastewaters from selected sites in each of the four towns. We went over the sampling collection and lab analysis in our previous study [14,15]. All study protocols, including primer sequences, can be found at https://www.protocols.io/workspaces/typhoides.

## Measurement of outcome and independent variables

**Outcome variable:** The dependent variable was a positive or negative answer to whether or not S. Typhi was found in wastewater samples. A positive outcome was one in which all three targets were met.

**Independent variables:** We used an Aquaprobe Model AP-2000 to measure the physicochemical parameters at each sampling site. This device has sensors for temperature, pH, salinity, seawater specific gravity, dissolved oxygen, turbidity, electrical conductivity, and oxidation-reduction potential. We used a questionnaire in the field to find out the flow rate, width, and depth of the water source. Variable selection was based on biological plausibility, univariate association with the outcome (p < 0.20), literature, and variance inflation factor (VIF < 5) to avoid multicollinearity. Lagged variables (e.g., prior-month positivity) were created to capture temporal dependencies.

## Statistical modeling approaches

In this study, both the non-spatial and spatial models were fitted and compared to determine which one can better predict the presence of S. Typhi.

### Non-spatial modeling

Three non-spatial modeling techniques were fitted to predict the presence of S. Typhi in wastewater. These models are described as follows:

1. **Generalized Estimating Equations (GEE) Model - (Logistic Regression with Independence Correlation Structure)**

The GEE model can be expressed as:

$$\log \left( \frac{p_{ij}}{1 - p_{ij}} \right) = X_{ij}^T \beta$$

(1)

Where:

- $p_{ij} = P(S\_Typhi = 1)$ for observation $j$ at site $i$
- $X_{ij}$ is Vector of covariates for observation $ij$
- $\beta$ are Vector of regression coefficients
- The model accounts for within-site correlation through the GEE framework

2. **Generalized Linear Mixed Effects Model (GLMM)**

The mixed-effects model formulation is:

$$\log \left( \frac{p_{ij}}{1 - p_{ij}} \right) = X_{ij}\beta + b_i$$

(2)

Where:

- $b_i \sim N(0, \sigma_{b2})$ is the random intercept for site $i$

- All other terms are as in the GEE model

- The random effects account for within-site correlation

### 3. Random Forest Model

The Random Forest is an ensemble model that does not have a simple mathematical formula, but can be conceptually represented as:

$$\hat{y} = majority\ vote\ \left( \left\{ T_k(x) \right\}_{k=1}^{B} \right)$$

(3)

Where:

- $T_k(x)$ is the prediction from the $k$-th decision tree

- $B = 500$ is the number of trees (as specified by ntree$=500$)

- Each tree is built on a bootstrap sample of the training data

- At each split, $mtry = 3$ predictors are considered

### Spatial modeling

In this study, four spatial models are considered and are discussed below:

### 1. Spatial GLMM (spaMM)

A Generalized Linear Mixed Model (GLMM) with a Matern covariance structure to account for spatial autocorrelation.

$$\log\left(\frac{p_{ij}}{1-p_{ij}}\right) = x_i^T \beta + \mu_i$$

(4)

Where:

- $x_i$ is the vector of fixed- effect predictore at site $i$.

- $\beta$ is the vector of fixed-effect coefficients.

- $\mu_i$ is the random effect at site $i$, which accounts for spatial dependence.

The vector of random effects $\mathbf{u} = (\mu_1,\ \ldots, \mu_n)^T$ is assumed to follow a multivariate normal distribution with a Matern covariance structure.

### 2. GAM with smoothing in space

A Generalised Additive Model (GAM) that uses thin-plate splines to predict and a Gaussian process (GP) smoother to smooth out spatial coordinates.

$$log\left(\frac{p_{ij}}{1-p_{ij}}\right) = \beta_0 + \sum_{j=1}^{p} f_j(x_i) + f_{spat}(s_i)$$

(5)

where:

- $\beta_0$ is the intercept

- $f_j$ are the smooth functions (thin-plate splines) for the $p$ predators or covariates.
- $f_{spat}$ is a Gaussian process smoother over the spatial coordinates $s_i = (Longitude_i, \; Latitude_i)$.

### 3. Random Forest with Spatial Features

A Random Forest model that includes spatial coordinates (Longitude, Latitude) as predictors.

$$\hat{y} = majority\ vote\ \left( \left\{ T_k(x_i,\ Longitude_i,\ Latitude_i) \right\}_{k=1}^{300} \right) \tag{6}$$

Where:

- $x_i$ includes all covariates in the above models.
- Each tree $T_k$ is trained on a bootstrap sample with **mtry = 3**.
- Spatial coordinates are treated as additional predictors.

### 4. Bayesian-like Spatial GAM (GAM with GP prior)

A Bayesian GAM with penalized splines (tp) for predictors and a Gaussian process (GP) prior for spatial smoothing.

$$\log \left( \frac{p_{ij}}{1 - p_{ij}} \right) = \beta_0 + \sum_{j=1}^{p} f_j \left( x_{ij} \right) + f_{spat} \left( s_i \right)$$

Where:

- $f_j$ penalized splines with Bayesian priors.
- $f_{spat}$ is a spatial term
- $x_i$ includes all covariates in the above models.

## Data preparation

**Data preprocessing:** The dataset was a balanced panel that included all 27 months of data collected from each site. We changed the response variable (S. Typhi - presence/absence) into a binary number format and treated sampling site (SITE_ID) and season as categorical factors. We made a spatial object for spatial models using latitude and longitude coordinates (WGS84, EPSG:4326) and kept the original coordinates as numeric variables for spatial predictors.

**Variable standardization**: We centred and scaled all of the continuous environmental predictors (temperature, pH, dissolved oxygen, electrical conductivity, oxygen reduction potential, salinity, and catchment population) to have a mean of zero and a unit variance. This made sure that effect sizes could be compared across variables with different measurement units. Before fitting the model, this standardisation was done to make the numbers more stable and help them converge.

## Data partitioning

We used a stratified 5-fold cross-validation (CV) framework to evaluate model performance and mitigate overfitting. Folds were created by grouping all observations from the same sampling site (SITE_ID), ensuring that no site appeared in both training and test sets within the same fold. In each CV iteration, 80% of the data (4 folds) were used for training and 20% (1 fold) for testing. Model hyperparameters were tuned through grid search on the training folds. Performance metrics

(accuracy, sensitivity, specificity) were computed on each test fold and then averaged across folds, providing a robust estimate of model generalizability to new locations and minimizing the risk of overfitting.

### Identifying key risk factors using SHAP analysis

We used a Random Forest model with SHapley Additive exPlanations (SHAP) analysis to find and understand the main environmental and temporal factors that led to the presence of S. Typhi. The model used both current measurements and values from one month prior for key water quality parameters, including temperature, pH, and dissolved oxygen. It also used seasonal indicators (month, season) and site characteristics (depth, width, catchment population).

Before we started analysing, we: (1) created temporal lag features (1-month intervals) to account for delayed environmental effects, (2) standardised all continuous predictors to make sure that the feature importance scales were comparable, and (3) removed missing observations (less than 5% of the data) using complete-case analysis.

We trained an optimized Random Forest classifier (1000 trees, permutation-based importance) on the complete dataset using probability outputs to enable SHAP value computation. For computational efficiency while maintaining representativeness, SHAP values were calculated for a stratified random subset of 500 observations using 50 Monte Carlo simulations per observation.

The SHAP study gave us: (1) global feature importance 1, measuring how much each predictor adds to the model's predictions; (2) directionality effects, Showing whether higher values of each parameter made it more or less likely to find S. Typhi.

We used permutation importance tests and out-of-bag error estimation to assess the model's robustness. To guarantee consistent value interpretation across analyses, a uniform background dataset was used for all SHAP computations.

### Metrics for classification performance

The following metrics were used to compare the models:

1. **Precision:** Total percentage of accurate forecasts (both positive and negative cases)

2. **Sensitivity:** True positive rate: the capacity to accurately detect the presence of S. Typhi

3. **Specificity:** True negative rate - ability to correctly identify absence conditions

## Results and discussion

### Site characteristics

In (Table 1), the environmental sampling data from Agogo, Domeabra, Hwidiem, and Juansa show notable differences in wastewater dynamics over 27 months, with 40 sampling sites. Agogo had the highest number of sampling sites (35) and contributed the majority of samples, totaling 883. There were 838 people living in the median catchment area of Agogo. Domeabra and Juansa had only two sites each, but we still got many samples: 27 from Domeabra and 52 from Juansa. Hwidiem gave 26 samples from one site where the median population was 472 people.

On the sampling days, flow speed was predominantly slow at Agogo, with 93.87% of samples collected from sluggish flows. In contrast, most of the samples from Hwidiem had a higher proportion of fast flows (88.5%), while Domeabra and Juansa also experienced mostly slow flows (3.26% and 2.48%, respectively).

Wastewater depths were primarily shallow (<5 cm) across all locations, particularly in Agogo (92.9%) and Domeabra (3.48%). Juansa, however, showed a higher prevalence of medium depths (14.77%). Deep sewage (>50 cm) was rare, constituting only 2.55% of samples. Regarding channel width, most samples were taken from narrow channels (<1 meter), especially in Agogo (95.93%) and Domeabra (4.07%). Juansa had wider channels, with 26% exceeding 2 meters. About 60% of all samples were collected during the wet season, indicating consistent seasonal representation.

**Table 1. Environmental sampling parameters across different locations.**

| S/N | Variables | TOWNS | | | | Total |
|---|---|---|---|---|---|---|
| | | **AGOGO** | **DOMEABRA** | **HWIDIEM** | **JUANSA** | |
| 1 | Number of ES sites | 35 | 2 | 1 | 2 | 40 |
| 2 | Number of samples collected | 883 | 27 | 26 | 52 | 988 |
| 3 | Catchment population (Median, IQR) | 838 (615,1437) | 890 (890,890) | 472 (472,472) | 865 (865,943) | |
| 4 | Flow Speed on day of sampling [n(%)] | n=874 | n=27 | n=26 | n=52 | n=979 |
| | Fast | 154 (72.99) | 1 (0.47) | 23 (10.9) | 33 (15.64) | 211 (21.55) |
| | Slow | 720 (93.87) | 26 (3.26) | 3 (0.39) | 19 (2.48) | 768 (78.45) |
| 5 | Depth of wastewater/sewage on day of sampling [n(%)] | | | | | |
| | Shallow (<5 cm) | 641 (92.9) | 24 (3.48) | 12 (1.74) | 13 (1.88) | 690 (70.48) |
| | Medium (5–50 cm) | 209 (79.17) | 2 (0.76) | 14 (5.30) | 39 (14.77) | 264 (26.97) |
| | Deep (>50 cm) | 24 (100) | 0 | 0 | 0 | 25 (2.55) |
| 6 | Width of wastewater/sewage on day of sampling [n(%)] | | | | | |
| | Less than 1 meter | 565 (95.93) | 24 (4.07) | 0 | 0 | 589 (60.16) |
| | Between 1 meter to 2 meter | 272 (80.24) | 2 (0.59) | 26 (7.67) | 39 (11.50) | 340 (34.73) |
| | >2 meter | 37 (74.00) | 0 | 0 | 13 (26.00) | 50 (5.11) |
| 7 | Season[n(%)] on day of sampling | | | | | |
| | Wet | 530 (89.23) | 16 (2.69) | 16 (2.69) | 32 (5.39) | 595 (60.22) |
| | Dry | 353 (89.82) | 10 (2.54) | 10 (2.54) | 20 (5.09) | 393 (39.78) |

## Distribution of S. Typhi positive detection

(Table 2) presents an analysis of the detection rates of Salmonella Typhi across various towns and seasons, as well as their association with the HF183 marker status. Out of all the samples, 44.13% tested positive for S. Typhi and 55.87% tested negative. There were big differences between the towns. Hwidiem had the highest positivity rate at 57.69%, followed by Agogo at 45.64%. Domeabra had the lowest positivity rate at 15.38%.

There was a statistically significant difference between these two groups ($\chi^2=17.71$, $p<0.001$).

The wet season had a much higher percentage of positive detections (50.17%) than the dry season (35.11%) ($\chi^2=21.74$, $p<0.001$). The identification of S. Typhi was strongly associated with the HF183 marker. The positivity rate for samples that tested positive for the HF183 marker was 45.70%, whereas the rate for samples that tested negative for the marker was 26.58%. This association was significant ($p=0.001$, $\chi^2=10.78$).

Furthermore, the presence of S. Typhi was influenced by flow conditions. Compared to samples collected under slow-flowing conditions, which had a positivity rate of 38.85%, samples collected under fast-flowing conditions showed a higher rate of 64.45%. Significant correlations were also found between wastewater depth and channel width. Specifically, medium sewage was more frequently associated with S. Typhi detection (58.33%) compared to shallow (39.42%) or deep (>50 cm) depths. Wider channels were correlated with a higher positivity rate of 58.7% compared to narrower channels.

These findings indicate that S. Typhi contamination is strongly linked to site characteristics such as town location, seasonality, wastewater flow dynamics, and indicators of fecal contamination (the HF183 marker). This emphasizes the importance of sanitation infrastructure in controlling the spread of pathogens.

## Fluctuations in monthly S. Typhi positivity rates

A temporal analysis of the monthly positive rates of S. Typhi from June 2022 to September 2024 shows significant fluctuations (Fig 1). The observed variations, which include an elevated positive rate during the June-September period of 2022 (peak≈80%) and a decline to approximately 15% around December 2022, may instead be attributable to other factors.

**Table 2. Detection of S. Typhi across Town, Season, and HF183 Status.**

| S/N | Variables | S. Typhi detection | | Pearson chi2(p-value) |
|---|---|---|---|---|
| | | Negative[n(%)=552 (55.87)] | Positive[n(%)=436 (44.13] | |
| 1 | TOWN | | | |
| | Agogo | 480 (54.36) | 403 (45.64) | |
| | Domeabra | 23 (84.62) | 4 (15.38) | |
| | Hwidiem | 11 (42.31) | 15 (57.69) | 17.71 (<0.001) |
| | Juansa | 38 (73.08) | 14 (26.92) | |
| 2 | SEASON | | | |
| | Dry | 255 (64.89) | 138 (35.11) | |
| | Wet | 297 (49.83) | 298 (50.17) | 21.74 (<0.001) |
| 3 | HF183 | | | |
| | Negative | 58 (73.42) | 21 (26.58) | |
| | Positive | 494 (54.30) | 415 (45.70) | 10.78 (0.001) |
| | Flow Speed on day of sampling [n(%)] | | | |
| | Fast | 75 (35.55) | 136 (64.45) | 43.94 (<0.001) |
| | Slow | 469 (61.15) | 298 (38.85) | |
| | Depth of wastewater/sewage on day of sampling [n(%)] | | | |
| | Shallow (<5 cm) | 418 (60.58) | 272 (39.42) | 28.87 (<0.001) |
| | Medium (5–50 cm) | 110 (41.67) | 154 (58.33) | |
| | Deep (>50 cm) | 16 (66.67) | 8 (33.33) | |
| | Width of wastewater/sewage on day of sampling [n(%)] | | | |
| | Less than 1 meter | 378 (64.18) | 211 (35.82) | 45.91 (<0.001) |
| | Between 1 meter to 2 meter | 140 (41.30) | 199 (58.70) | |
| | >2 meter | 26 (52.00) | 24 (48.00) | |

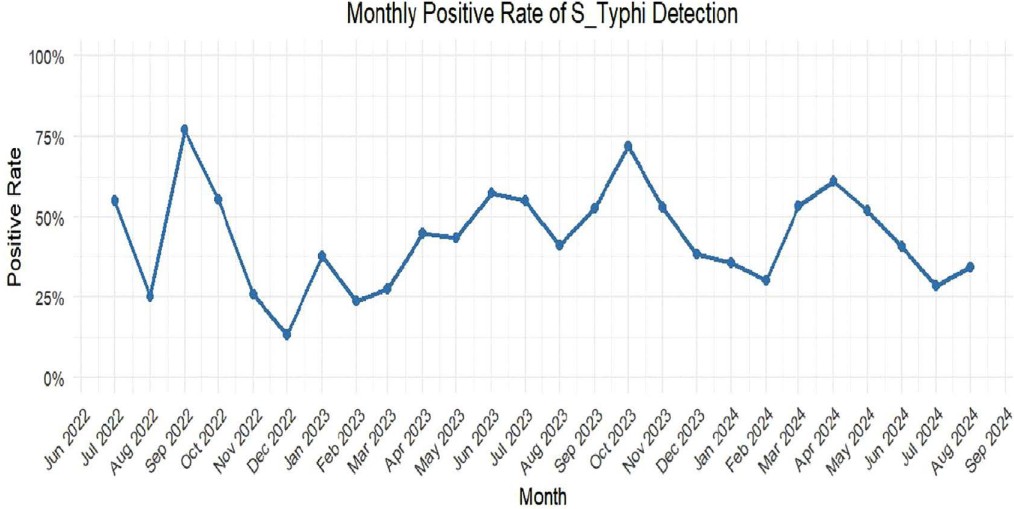

**Fig 1. A time series plot showing the monthly positive rate of S. Typhi (Salmonella Typhi) detection from June 2022 to September 2024.**

Mid-2023 saw a moderate increase (55–60%), while the trend in 2024 appears variable, ending with approximately 30% in September 2024.

### Association of wastewater quality parameters with Salmonella Typhi detection

(Table 3) shows that there are strong links between the detection of S. Typhi in wastewater and specific water quality parameters.

Positive samples had higher pH levels (7.46 vs. 7.40, p < 0.001) and dissolved oxygen concentrations (46.97 vs. 36.77 mg/L, p < 0.001). We also observed a significant decrease in total dissolved solids in positive samples (1092.2 vs. 1172.35 mg/L, p = 0.048). This could mean that the samples were diluted or that they interacted with organic matter.

There were no significant links between temperature, oxygen reduction potential, electrical conductivity, salinity, or seawater specific gravity (p > 0.05). There was also a significant difference in rainfall between the positive and negative samples (3.92 mm vs. 3.30 mm, p = 0.022), suggesting that factors such as rainfall may influence the presence of S. Typhi in water.

### Choosing non-spatial models

(Table 4) shows how well three temporal models did at predicting the presence of S. Typhi in wastewater samples. The Generalised Estimating Equations (GEE) model did not perform well (Sensitivity: 0.352, Specificity: 0.861, Accuracy: 0.608), indicating that it is not suitable for making reliable predictions.

The Mixed-Effects model showed slight improvement (Sensitivity: 0.493, Specificity: 0.750, Accuracy: 0.622), but it still lacked the required accuracy for public health use. In contrast, the Random Forest model performed exceptionally well, accurately predicting both positive and negative cases (Sensitivity: 0.997, Specificity: 0.989, Accuracy: 0.993). This

**Table 3. Mean differences in water quality parameters by S. Typhi detection status (with 95% CIs).**

| Parameter | S Typhi Detection | | P_Value* |
|---|---|---|---|
| | **Negative Mean(C1)** | **Positive Mean(C1)** | |
| Temperature (°C) | 27.46 (27.16, 27.76) | 27.42 (27.13, 27.71) | 0.87 |
| Oxygen_reduction_potential(mV) | 38.01 (−110.03, 186.05) | 5.13 (−127.01, 137.27) | 0.11 |
| pH | 7.4 (6.92, 7.88) | 7.46 (7.26, 7.66) | <0.001 |
| dissolved_oxygen (%) | 36.77 (29.26, 44.28) | 46.97 (39.89, 54.05) | <0.001 |
| Electrical_conductivity (µs/cm) | 2037.47 (1815.59, 2259.34) | 1899.04 (1648.46, 2149.62) | 0.12 |
| Total_dissolved_solids (mg/L) | 1172.35 (1080.31, 1264.4) | 1092.2 (979.82, 1204.58) | 0.048 |
| Salinity (psu) | 1.57 (0.34, 2.79) | 0.88 (0.79, 0.97) | 0.14 |
| Seawater_specific_gravity (st) | 1.13 (−0.44, 2.69) | 0 (0, 0) | 0.67 |
| Rainfall (mm) | 3.30(2.96,3.65 | 3.92(3.51,4.33 | 0.022 |

*\* The p-values were obtained using the Two-sample Wilcoxon rank-sum (Mann-Whitney) test.*

**Table 4. Comparison of performance metrics of competing non-spatial models.**

| Model | Sensitivity | Specificity | Accuracy |
|---|---|---|---|
| GEE Model (Logistic) | 0.352 | 0.861 | 0.608 |
| Mixed-Effects Model | 0.493 | 0.750 | 0.622 |
| Random Forest | 0.997 | 0.989 | 0.993 |

suggests that the Random Forest model is better able to capture the complex, non-linear patterns found in the temporal data of S. Typhi.

## Selection of spatial models

(Table 5) provides a comparison of four models for the spatial prediction of S. Typhi presence. There is a Bayesian Spatial GAM, a GAM with Spatial Smoothing, a Random Forest that includes spatial features, and a Spatial GLMM that is implemented through spaMM. The accuracy of these models ranged from 0.650 to 0.688. The Random Forest model achieved the highest accuracy of 0.688, indicating that it was more effective in overall prediction.

The sensitivity values, which show how well the models can find true positives, were between 0.547 and 0.596 for the Random Forest, and 0.600 for the Spatial GLMM. The Bayesian Spatial GAM and the GAM with Spatial Smoothing had the same value.

The Random Forest had the highest specificity of 0.767, and the Bayesian Spatial GAM had the lowest of 0.734. Overall, the Random Forest with spatial features had the best balance of accuracy, sensitivity, and specificity among the models tested. This means that it is likely to give the best predictive performance for this application.

## Comparing spatial and temporal models

Non-spatial models, particularly the Random Forest algorithm, performed significantly better than spatial models, achieving high accuracy (0.993), sensitivity (0.997), and specificity (0.989). We observed that the spatial models performed less effectively, with the best model, Random Forest with spatial smoothing, achieving moderate accuracy (0.688) and specificity (0.767), but a lower sensitivity of 0.596. Based on these findings, we recommend using the non-spatial Random Forest model, as it demonstrates better predictive performance.

## Using the non-spatial random forest model to identify important environmental and time-related risk factors

(Fig 2) shows the most important predictors found by SHAP analysis, with the top five risk factors highlighted. Among these, pH has the greatest effect on the S. Typhi detection rate, as indicated by its long SHAP bar, which suggests that acidic conditions increase the likelihood of finding S. Typhi. The time of year is also very important. The wet season makes detection much more likely, which is consistent with higher positivity rates during times of more runoff and contamination (50.17% in the wet season vs. 35.11% in the dry season). Dissolved oxygen (DO) is another important predictor. Low DO levels increase the likelihood of detection because S. Typhi grows better in anaerobic conditions. This is evident in the higher prevalence in samples with low DO (46.97%) compared to the negatives (36.77%).

Prior-month positivity (S_Typhi_lag1) is a strong temporal predictor. This indicates that past detection patterns can predict future outbreaks, with clear patterns of spread, such as the mid-2023 surge following lows in December 2022. Channel width, catchment population, and flow speed are other factors that have a moderate effect on risk. For example, intermediate channel widths (1–2 meters) are linked to higher positivity, possibly because they create stagnation zones. On the other hand, very narrow or wide channels are generally safer.

**Table 5. A comparison of the performance metrics of different spatial models.**

| Model | Accuracy | Sensitivity | Specificity |
| --- | --- | --- | --- |
| Bayesian Spatial GAM | 0.650 | 0.547 | 0.734 |
| GAM with Spatial Smoothing | 0.651 | 0.547 | 0.734 |
| Random Forest with spatial smoothing | 0.688 | 0.596 | 0.767 |
| Spatial GLMM (spaMM) | 0.679 | 0.600 | 0.747 |

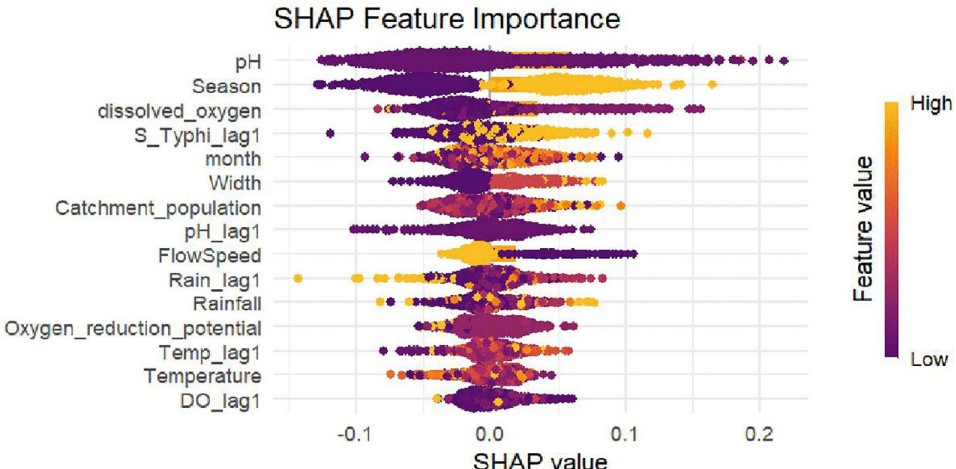

**Fig 2. SHAP feature importance for *S. Typhi* detection.**

(Fig 3) delves into the directionality of these relationships. Low pH (acidic water) sharply increases risk, supporting mechanisms where acidity supports bacterial survival, whereas high pH diminishes it. Seasonally, the wet period markedly elevates S. Typhi detection, consistent with increased runoff and contamination. Low dissolved oxygen makes it easier for bacteria to grow, while high oxygen levels protect against this. The flow speed has a non-linear relationship: both very slow and very fast flows are linked to higher detection rates. Stagnation concentrates bacteria, and rapid spread spreads contamination. Channel width also affects the detection rate. Intermediate widths (1–2 meters) are most likely to be positive, while narrower (<1m) or wider (>2m) channels have a lower detection rate, which is consistent with what is seen in nature. The strong predictive relevance of prior-month positivity emphasizes outbreak self-propagation, highlighting the importance of temporal monitoring.

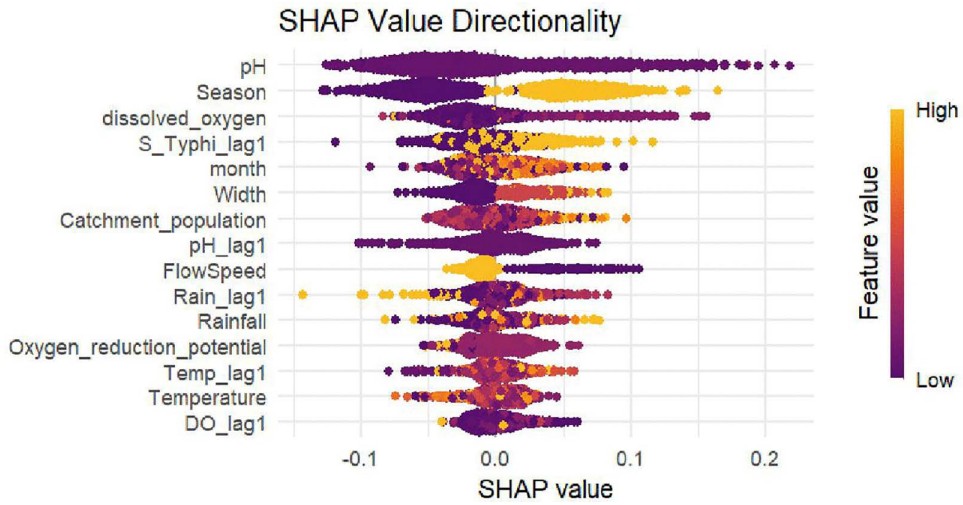

**Fig 3. SHAP value directionality for *S. Typhi* detection.**

## Discussion

This study analyzes S. Typhi prevalence in wastewater at four locations in Ghana, highlighting the environmental and temporal factors in typhoid transmission.

### Key environmental drivers of S. Typhi detection

Our findings confirm that wastewater characteristics profoundly impact S. Typhi prevalence. The strong link between HF183, a marker for human faeces ($\chi^2 = 10.78$, p = 0.001; Table 2), and contamination shows that human sewage inputs are the source of the contamination. This makes HF183 a good way to detect typhoid in wastewater-based epidemiology [14,16]. Flow speed emerged as a critical factor, with fast-flowing wastewater exhibiting significantly higher positivity rates (64.45%) compared to slow flows (38.85%; p < 0.001). Conversely, intermediate channel widths (1–2 meters) and medium depths (5–50 cm) were associated with the highest positivity (58.70% and 58.33%, respectively; Table 2), suggesting that these conditions foster bacterial accumulation or persistence, potentially due to reduced dilution or stagnation zones [17]. The significant seasonal pattern, with higher positivity during wet seasons (50.17% vs. 35.11% during dry seasons; p < 0.001), aligns with known typhoid epidemiology in endemic regions and likely reflects increased runoff contaminating channels and/or reduced wastewater dilution [14,18–21].

### Water quality interactions

Water chemistry parameters further elucidated the ecology of S. Typhi. Higher pH (7.46 vs. 7.40 in negative samples; p < 0.001) and dissolved oxygen (46.97% vs. 36.77%; p < 0.001) in positive samples (Table 3) suggest that S. Typhi may thrive in less acidic, oxygen-rich environments; however, further research is needed to clarify the underlying mechanisms. The association with higher rainfall (3.92 mm vs. 3.30 mm; p = 0.022) supports the seasonal findings, implicating precipitation in pathogen mobilization. Lower total dissolved solids (TDS) in positive samples (1092.2 vs. 1172.35 mg/L; p = 0.048) suggest possible dilution effects or interactions with organic particulates that may influence detection.

### Geographical and temporal heterogeneity

Significant inter-town variation ($\chi^2 = 17.71$, p < 0.001; Table 2) highlights localized detection factors. Hwidiem's high positivity (57.69%) despite fewer sites warrants investigation into local sanitation infrastructure or population density effects. The significant fluctuations over time, shown in (Fig 1), highlight the variable nature of environmental transmission, with peaks around 80% and troughs near 15%. The predictive power of prior-month positivity (S_Typhi_lag1) in the Random Forest model (Figs 2 and 3) confirms temporal autocorrelation, suggesting outbreak propagation or persistent environmental reservoirs.

### Model performance and predictive insights

A critical finding is the superior performance of the temporal Random Forest model (Accuracy: 0.993, Sensitivity: 0.997, Specificity: 0.989; Table 4) over spatial models (best spatial Random Forest Accuracy: 0.688; Table 5) and traditional statistical models (GEE, Mixed-Effects) [22,23]. This demonstrates that temporal patterns (seasonality, historical positivity) and non-linear interactions among environmental variables are crucial for predicting S. Typhi detection, patterns that are effectively captured by machine learning but often missed by linear or spatial-only approaches [24]. However, the exceptionally high metrics require caution due to the risk of overfitting. We addressed this concern through site-stratified cross-validation, hyperparameter tuning, and out-of-bag error estimation. The model's performance remained consistent across folds, demonstrating its robustness. The SHAP analysis identified the dominant role of pH (low values increasing detection), season (wet season high detection), and dissolved oxygen (low DO increasing detection) (Figs 2 and 3), aligning with our univariate results. The complex, non-linear relationships revealed, such as U-shaped effects for flow speed

(both very slow and very fast flows increasing detection) and channel width (intermediate widths increasing detection), highlight the necessity of advanced modeling to unravel environmental pathogen dynamics [25].

## Public health implications

The findings underscore several key public health implications. First, seasonality plays a crucial role, highlighting the importance of intensifying typhoid surveillance and preventive measures prior to and during the wet season when environmental conditions favor transmission. Additionally, local hydrological features warrant targeted interventions; specifically, channels with intermediate widths (1–2 meters) and medium depths should be prioritized for remediation efforts, especially during periods of fluctuating flow conditions that may promote pathogen persistence and spread. The strong association with the HF183 marker emphasizes that improving sewage containment and treatment infrastructure is essential in reducing environmental contamination and subsequent infection risk. Finally, integrating advanced temporal machine learning models, such as Random Forest, with real-time inputs, including pH, rainfall, dissolved oxygen, and historical positivity data, can enhance outbreak prediction and facilitate proactive public health responses.

## Limitations and future research

The study has several limitations: (1) uneven sampling across towns may affect the generalizability of the results; (2) while the high model accuracy raises concerns about overfitting, this issue has been addressed through rigorous validation; and (3) the HF183 marker does not differentiate between typhoid carriers. Future research should aim to validate the model in other regions, incorporate genomic data, and explore real-time integration with public health informatics systems.

## Conclusion

This study demonstrates that Salmonella enterica serovar Typhi detection in wastewater is driven by synergistic environmental, temporal, and spatial factors. Key predictors include low pH, wet season conditions, reduced dissolved oxygen, intermediate channel widths (1–2 meters), and elevated flow speeds, all significantly elevating detection rate. The strong association with the human fecal marker (HF183) confirms sewage contamination as a primary source. Critically, temporal dynamics, particularly historical positivity and seasonal rainfall, outweighed spatial factors in predictive power. The Random Forest model, utilizing temporal data, achieved a high accuracy of 99.3%, significantly surpassing both spatial and traditional statistical models. This highlights the importance of non-linear, time-dependent interactions in the transmission of environmental S. Typhi. This model holds promise for practical implementation in public health surveillance networks, such as the Global Health Security Agenda (GHSA) and WHO GLASS, enabling proactive, environment-based typhoid monitoring in endemic regions.

## Author contributions

**Conceptualization:** Sampson Twumasi-Ankrah, Michael Owusu, Nicholas C. Grassly, Yaw Adu-Sarkodie, Ellis Owusu-Dabo.

**Data curation:** Sampson Twumasi-AnkraH, Seidu Amenyaglo, Caleb Osei-Wusu Sarfo, Portia Okyere Boakye, Christopher B. Uzzell.

**Formal analysis:** Sampson Twumasi-Ankrah.

**Funding acquisition:** Nicholas C. Grassly.

**Investigation:** Sampson Twumasi-Ankrah, Michael Owusu, Michael Owusu-Ansah, Eric Darko, Christopher B. Uzzell.

**Methodology:** Sampson Twumasi-Ankrah, Michael Owusu, Michael Owusu-Ansah, Eric Darko, Christopher B. Uzzell, Isobel M. Blake, Nicholas C. Grassly.

Resources: Nicholas C. Grassly.

Software: Sampson Twumasi-Ankrah.

Supervision: Nicholas C. Grassly, Yaw Adu-Sarkodie, Ellis Owusu-Dabo.

Validation: Michael Owusu, IsobeL M. Blake, Nicholas C. Grassly, YAW Adu-Sarkodie, Ellis Owusu-Dabo.

Visualization: Sampson Twumasi-Ankrah, Christopher B. Uzzell.

Writing – original draft: Sampson Twumasi-Ankrah.

Writing – review & editing: Sampson Twumasi-Ankrah, Michael Owusu, Michael Owusu-Ansah, Seidu Amenyaglo, Caleb Osei-Wusu Sarfo, Eric Darko, Portia Okyere Boakye, Christopher B. Uzzell, Isobel M. Blake, Nicholas C. Grassly, Yaw Adu-Sarkodie, Ellis Owusu-Dabo.

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
