## [Decision Letter · Decision Letter 0]

10 Dec 2025

Response to Reviewers
Revised Manuscript with Track Changes
Manuscript

Shaden Kamhawi

co-Editor-in-Chief

Paul Brindley

co-Editor-in-Chief

**Journal Requirements:**

1) Please provide an Author Summary. This should appear in your manuscript between the Abstract (if applicable) and the Introduction, and should be 150-200 words long. The aim should be to make your findings accessible to a wide audience that includes both scientists and non-scientists. Sample summaries can be found on our website under Submission Guidelines:

3) Please amend your detailed Financial Disclosure statement. This is published with the article. It must therefore be completed in full sentences and contain the exact wording you wish to be published.

4) Please revise your current Competing Interest statement to the standard "The authors have declared that no competing interests exist."

5) We noted that you have cited tables (1 ,2) , and then tables ( from 5 to 7)  ; however, tables 3 and 4 have not been cited or included in the manuscript. Please re-label and number the tables in a numerical order.

**Reviewers' comments:**

**Key Review Criteria Required for Acceptance?**

**Methods**

-Are the objectives of the study clearly articulated with a clear testable hypothesis stated?

-Is the study design appropriate to address the stated objectives?

-Is the population clearly described and appropriate for the hypothesis being tested?

-Is the sample size sufficient to ensure adequate power to address the hypothesis being tested?

-Were correct statistical analysis used to support conclusions?

-Are there concerns about ethical or regulatory requirements being met?

Reviewer #1: The applied methods are consistent with the stated objectives of the study.

Reviewer #2: (No Response)

**Results**

-Does the analysis presented match the analysis plan?

-Are the results clearly and completely presented?

-Are the figures (Tables, Images) of sufficient quality for clarity?

Reviewer #1: The results correspond to the analysis plan and are sufficiently presented.

Reviewer #2: (No Response)

**Conclusions**

-Are the conclusions supported by the data presented?

-Are the limitations of analysis clearly described?

-Do the authors discuss how these data can be helpful to advance our understanding of the topic under study?

-Is public health relevance addressed?

Reviewer #1: The conclusions are supported by the conducted work and obtained data.

Reviewer #2: (No Response)

**Editorial and Data Presentation Modifications?**

Reviewer #1: (No Response)

Reviewer #2: (No Response)

**Summary and General Comments**

Reviewer #1: 1. Overall Evaluation

The manuscript presents an original and relevant study focused on the application of statistical and machine learning models to predict the presence of Salmonella Typhi in wastewater samples from Ghana. The work is methodologically sound, well-structured, and addresses a significant public health issue related to wastewater-based surveillance (WBE). The authors combine classical regression approaches (GLMM, GAM, GEE) with modern non-parametric algorithms (Random Forest, SHAP), providing a comprehensive comparative analysis. The findings confirm the efficiency of non-parametric models and highlight the environmental and seasonal factors influencing S. Typhi persistence.

2. Strengths

- High scientific and practical relevance for WBE systems in endemic regions.

- Thorough data processing and the use of several complementary analytical methods.

- Clear structure, well-presented results, and consistent discussion with literature.

- Good graphical and tabular quality, making the findings easy to interpret.

3. General Recommendations

The manuscript appears complete and well-prepared. To further enhance transparency and reproducibility, it is recommended to:

- Clarify methodological details such as variable selection criteria, model validation parameters, and data partitioning into training/testing subsets.

- Expand the discussion of study limitations, including sample size, seasonality, and potential overfitting concerns given the high accuracy metrics (Accuracy > 0.99).

- Slightly reinforce the interpretation of ecological and biological mechanisms underlying S. Typhi persistence in environmental contexts.

- In the Conclusion, emphasize the potential contribution of the model to practical public health systems (GHSA, WHO GLASS).

4. Significance and Impact

This study has strong potential for application in endemic regions and can serve as a methodological model for similar pathogen monitoring efforts (e.g., Vibrio cholerae, Salmonella Paratyphi). It expands the methodological foundation for early warning systems based on wastewater surveillance and contributes to regional biosecurity capacities.

5. Overall Recommendation

Minor revision. The manuscript is suitable for publication after minor textual and methodological clarifications.

Reviewer #2: (No Response)

PLOS authors have the option to publish the peer review history of their article (what does this mean? ). If published, this will include your full peer review and any attached files.

**Do you want your identity to be public for this peer review?** For information about this choice, including consent withdrawal, please see our Privacy Policy .

Reviewer #1: No

Reviewer #2: **Yes:** Manfred Dakorah Asiedu

**Figure resubmission:**

**Reproducibility:** To enhance the reproducibility of your results, we recommend that authors of applicable studies deposit laboratory protocols in protocols.io, where a protocol can be assigned its own identifier (DOI) such that it can be cited independently in the future. Additionally, PLOS ONE offers an option to publish peer-reviewed clinical study protocols. Read more information on sharing protocols at https://plos.org/protocols?utm_medium=editorial-email&utm_source=authorletters&utm_campaign=protocols

---

## [Editor Report · Decision Letter 1]

27 Jan 2026

Dear Dr. Twumasi-Ankrah,

We are pleased to inform you that your manuscript 'Statistical Methods for Predicting the Presence of Salmonella Typhi in Wastewater Samples at Asante Akyem Agogo, Ghana.' has been provisionally accepted for publication in PLOS Neglected Tropical Diseases.

Best regards,

Travis J Bourret

Academic Editor

Ana LTO Nascimento

Section Editor

Shaden Kamhawi

co-Editor-in-Chief

Paul Brindley

co-Editor-in-Chief

---

## [Editor Report · Acceptance letter]

Dear PROF Twumasi-Ankrah,

We are delighted to inform you that your manuscript, "Statistical Methods for Predicting the Presence of Salmonella Typhi in Wastewater Samples at Asante Akyem Agogo, Ghana.," has been formally accepted for publication in PLOS Neglected Tropical Diseases.

Best regards,

Shaden Kamhawi

co-Editor-in-Chief

Paul Brindley

co-Editor-in-Chief
